# Endogenous Microbacteria Can Contribute to Ovarian Carcinogenesis by Reducing Iron Concentration in Cysts: A Pilot Study

**DOI:** 10.3390/microorganisms12030538

**Published:** 2024-03-07

**Authors:** Naoki Kawahara, Shoichiro Yamanaka, Kyohei Nishikawa, Motoki Matsuoka, Tomoka Maehana, Ryuji Kawaguchi, Naoki Ozu, Tomomi Fujii, Aya Sugimoto, Akihiko Yoshizawa, Fuminori Kimura

**Affiliations:** 1Department of Obstetrics and Gynecology, Nara Medical University, Kashihara 634-8522, Japan; shoichiroyamanaka@naramed-u.ac.jp (S.Y.); k196560@naramed-u.ac.jp (K.N.); kawaryu@naramed-u.ac.jp (R.K.); kimurafu@naramed-u.ac.jp (F.K.); 2Institute for Clinical and Translational Science, Nara Medical University Hospital, Kashihara 634-8522, Japan; nao.oz@naramed-u.ac.jp; 3Department of Transformative System for Medical Information Osaka, University Graduate School of Medicine, Osaka University, Osaka 565-0871, Japan; fujit@cider.osaka-u.ac.jp; 4Division of Fostering Required Medical Human Resources, Center for Infectious Disease Education and Research (CiDER), Osaka University, Osaka 565-0871, Japan; 5Department of Diagnostic Pathology, Nara Medical University, Kashihara 634-8522, Japan; a_sugimoto@naramed-u.ac.jp (A.S.); akyoshi@naramed-u.ac.jp (A.Y.)

**Keywords:** endometriosis-associated ovarian cancer, ovarian endometrioma, iron, *Bacillus*, *Propionibacterium*, *Lactobacillus_perolens*, *Acetobacter_aceti*

## Abstract

Among epithelial ovarian cancer, clear cell carcinoma is common for chemo-resistance and high mortality. This cancer arises from benign ovarian endometrioma (OE), which is a high oxidative stress environment due to the cystic retention of menstrual blood produced during menstruation and the “iron” liberated from the cyst. There has been strong evidence that the iron concentration in OE decreases when they become cancerous. A decrease in iron concentration is a necessary condition for the formation of cancer. However, the mechanism of carcinogenesis is not yet clear. In the current study, the bacterial flora in endometriosis-associated ovarian cancer (EAOC), including clear cell carcinoma, and their origin, OE, were investigated using next-generation sequencing. The Shannon index in the genus level was significantly higher in EAOC than in OE fluids. Among several bacterial flora that were more abundant than benign chocolate cysts, a number of bacterial species that correlate very well with iron concentrations in the cysts were identified. These bacterial species are likely to be associated with decreased iron concentrations and cancer development.

## 1. Introduction

Epithelial ovarian cancer is the fifth leading cause of cancer-related death in women in the USA [1], and about 200,000 new cases are reported annually worldwide [2,3]. Ovarian cancer is divided into epithelial, germ cell, and sex cord-stromal tumors, as well as epithelial ovarian cancer, which has the highest rate at over 90% [4,5]. The major subtypes of ovarian tumors are clear cell carcinoma (CCC), endometrioid carcinoma (EC), high-grade serous carcinoma (HGSC), and mucinous carcinoma. CCC and EC are named endometriosis-associated ovarian cancer (EAOC), which are a well-known type arising from benign endometrioma [6,7]. EAOC accounts for about 40% of epithelial ovarian cancer (EOC) in Japan, and they are more common than in Europe and the USA [8,9,10,11,12].

EAOC is considered a stress-tolerant cancer due to its origin from chocolate cysts in a highly oxidative microenvironment induced by accumulative excess heme-iron released from repeated menstrual bleeding, which causes chronic inflammation. Yamaguchi et al. first reported in 2008 that the contents of the endometriotic cysts were rich in free iron and under excessive oxidative stress [13]. Elevated levels of reactive oxygen species (ROS) can stimulate the growth of endometriotic lesions [14], and then endometriotic cells receive sublethal damage which may lead to carcinogenesis [10,15]. Yoshimoto et al. conducted a detailed study on the levels of iron-related compounds in cyst fluids of benign OE and EAOC. Interestingly, the concentration of total iron, heme iron, and free iron was found to be significantly higher in OE cyst fluid than in EAOC cysts [16]. The above research suggests that mild oxidative stress is a more likely cause of carcinogenesis than excessive iron exposure. There is a hypothesis that reducing iron may promote cancer, but no conclusive evidence was found to support this claim.

There has been growing interest in the interaction between microbiota and malignancies. For example, H. pylori was proven to target different cellular proteins to modulate the host inflammatory response and initiate multiple “hits” on the gastric mucosa, resulting in chronic gastritis and peptic ulceration and leading to gastric cancer or mucosa-associated lymphoid tissue (MALT) lymphoma [17]. Although the bacteria responsible for ovarian cancer carcinogenesis have not yet been identified, Nejman et al. reported some microbiome was found in tumor tissue. Still, they concluded that their data do not establish whether intra-tumor bacteria play a causal role in the development of cancer or whether their presence simply reflects infections of established tumors [18]. In this study, we analyzed the microbiome in cyst fluid and compared its proportions between EAOC and OE.

## 2. Materials and Methods

### 2.1. Patients

We compiled a list of patients who had previously untreated, histologically confirmed ovarian tumors and were treated at Nara Medical University Hospital from January 2013 to June 2022. This study included cases of benign ovarian tumors and malignant tumors. Patients over 20 years old at the time of surgery and receiving magnetic resonance imaging (MRI) after hospitalization were included in the cohort. Patients who were under 20 years old, prone to claustrophobia, or contraindicated for MRI were excluded. All cases were histologically confirmed, and written consent was obtained for the use of patients’ clinical data for research. A total of 57 patients were included in the study, with 34 being benign OE cases and 23 being malignant cases. No patients underwent chemotherapy or radiotherapy for their ovarian tumors before treatment. We collected data on age, body mass index (BMI), parity, postoperative diagnosis, including FIGO (The International Federation of Gynecology and Obstetrics) stage, and MRI results, including tumor diameter and R2 value, from the patient’s medical records.

### 2.2. DNA Extraction and Bacterial 16S rDNA Sequencing from Cyst Fluid Samples

The samples were frozen and kept at a temperature of −80 °C. When they were dissolved at 37 °C, they were mixed with phosphate-buffered saline (PBS) ten times and then briefly vortexed. After that, the samples were stored at −80 °C until the 16S rRNA NGS process. QIAamp DNA Microbiome Kit (#51704, QIAGEN, Venlo, The Netherlands) was used to extract bacterial genomic DNA based on the manufacturer’s instructions. The 16s ribosomal DNA (rDNA) V4 region was amplified using primers 515 F (*5′-TCGTCGGCAGCGTCAGATGTGTATAAGAGACAGGTGYCAGCMGCCGCGGTAA-3′*) and 806 R (*5′-GTCTCGTGGGCTCGGAGATGTGTATAAGAGACAGGGACTACNVGGGTWTCTAAT-3′*) with KAPA HiFi HotStart ReadyMix (2X) (# KK2601, KAPA Biosystems, Wilmington, MA, USA) and Nextera XT v2 Index Kit (# FC-131-2001 [Set A]-# FC-131-2004 [Set D], Illumina, Inc., San Diego, CA, USA) as an index primer for 2ndPCR. The library quality check was conducted by Tapestation4200 (Agilent, Santa Clara, CA, USA) with Genomic DNA Reagents (#5067-5366, Agilent, Santa Clara, CA, USA), Genomic DNA Screen Tape (#5067-5365, Agilent, Santa Clara, CA, USA), and Loading Tip (#5067-5598, Agilent, Santa Clara, CA, USA). The PCR products were sequenced on the Illumina Miniseq (Illumina, Inc., San Diego, CA, USA) with MiniSeq Mid Output Kit (#FC-420-1004, Illumina, Inc., San Diego, CA, USA) and NextSeq PhiX Control Kit (FC-110-3002, Illumina, Inc., San Diego, CA, USA). Sequencing analysis was subsequently performed by Illumina BaseSpace 16S Metagenomics App (Version 1.1.0).

### 2.3. Tumor Imaging and Diagnoses

Patients were first examined at the outpatient clinic, where they underwent an internal examination, which included an ultrasound. The next step involved routine MR imaging using T1W and T2W sequences to determine the tumor’s largest diameter among axial, sagittal, and coronal imaging. This measurement was then used to calculate the tumor’s volume (D1 × D2 × D3 × π/6) from the three diameters (D1, D2, and D3). The patients were initially diagnosed with OE or EAOC through MRI, which was later confirmed by a histological examination of surgically removed tissue. This examination was conducted by at least two pathologists who were unaware of the study. The R2 values were obtained through a 3T system, Magnetom Verio or Skyla, by Siemens Healthcare, based in Erlangen, Germany. After the clinical MR imaging, the registered patients underwent MR relaxometry using the single-voxel acquisition mode sequence at multiple echo times. An exponential decay was then fitted to the echo amplitude at different multiple echo times [19]. A parameter R2 value (s^−1^) was calculated using a high-speed T2 *-corrected multi-echo MR sequence (HISTO) by the 3T–MR system in vivo and ex vivo, which was previously described [20,21]. The HISTO sequence was based on the single-voxel steam sequences that could be used for relative fat quantification in the liver [22]. This method can estimate the amount of iron stored in the liver by measuring the T2 of water, which changes according to iron concentration. Using an imaging platform from Siemens Medical Systems in Erlangen, Germany, we designed and programmed a pulse sequence for use with a 3T system. To analyze the cyst fluid, five measurements are taken at different TEs: 12, 24, 36, 48, and 72 ms. While holding the patient’s breath, the standard procedure is performed and takes 15 s. For signal saturation prevention and acceptable acquisition time, the repetition time (TR) is fixed at 3000 ms. A voxel (VOI) of 30 × 30 × 30 mm spectroscopy is used, which is positioned in the center of the OE or EAOC cyst by a female pelvic MR imaging specialist radiologist. The VOI is placed only to select the liquid portion of the cyst, not the solid portion. If a patient has more than one cyst, the fluid from the largest one is measured.

### 2.4. Statistical Analysis

We used SPSS version 25.0 (IBM SPSS, Armonk, NY, USA) and RStudio (2023.03.0+386) for our analyses. We calculated the Shannon index and conducted a Mann–Whitney U test to compare the DNA copy number proportion between benign and malignant tumor cyst fluid. We also calculated the Jaccard index and performed a principal component analysis. Additionally, we performed a multiple regression analysis to determine the correlation between the degree of abundance of bacterial genus or species and the R2 value. We considered a two-sided *p*-value of less than 0.05 to be statistically significant.

## 3. Results

### 3.1. Patients

From January 2013 to June 2022, a total of 57 patients were included in this study. The demographic and clinical characteristics of the current cohort are outlined in Table 1. Age, tumor volume, and R2 value showed significant differentiation.

### 3.2. The Taxonomic Diversity of Cyst Fluid Microbiota

Firstly, we calculated the Shannon index, which indicates the diversity of a single sample. In other words, it is a sample-specific index; the higher the value, the higher the species diversity. The Shannon index in genus level, which measures species richness and evenness, was significantly higher in EAOC than in OE fluids (1.75 [0.03–2.21] vs. 1.84 [1.51–2.91], *p* = 0.047, respectively) (Figure 1a). However, the Shannon index at the species level did not show significant differentiation between these groups (1.91 [0.05–2.35] vs. 2.03 [1.65–2.61], *p* = 0.057, respectively). We used the Jaccard index to measure the diversity difference between the two samples. The index measures the distance between two points, where a greater distance indicates a more significant difference in the compositions of the two samples. At both genus and species levels, the ß diversity between the EAOC and OE fluids showed that the EAOC group was considerably different from that of the EAOC (as shown in Figure 1b).

We hypothesized that if certain bacteria dominate the malignant cyst fluid, it could induce malignancy in epithelial cells. We conducted a study to examine whether bacteria in the tumor cyst fluid have a significant impact on malignant transformation by analyzing the expression rate of bacteria in EAOC and OE cyst fluids. As seen in Figure 2a,b, the heat maps for genus and species were found to be markedly different between the groups. Specifically, we investigated the specific bacteria whose expression rate occupied over 50% of the bacterial flora.

### 3.3. Isolation of Key Bacteria in Cyst Fluid

Since the cyst fluid microbiota of EAOC has a higher alpha and beta diversity, the mapping provided an important key to elucidate the specific EOAC-related bacteria. Candidates had at least 50% expression and a median occupancy significantly higher in EAOC than in OE. At genus level, *Acetobacter*, *Lactobacillus*, *Propionibacterium*, *Bacillus*, *Pseudomonas*, *Gluconobacter*, *Acidomonas*, *Lysinibacillus*, *Komagataeibacter*, *Chryseobacterium*, *Rhizobium*, *Gluconacetobacter*, *Halotalea*, *Clostridium_sensu_stricto*, *Paracoccus*, *Swaminathania*, *Arcobacter*, *Escherichia/Shigella*, *Xanthomonas*, *Clostridium_XlVa*, *Streptococcus*, *Enterococcus*, and *Azomonas* were extracted (Figure 3a). At species level, *Acidomonas_methanolica*, *Gluconacetobacter_liquefaciens*, *Halotalea_alkalilenta*, *Lysinibacillus_sp.*, *Lactobacillus_perolens*, *Acetobacter_tropicalis*, *Lactobacillus_vini*, *Swaminathania_salitolerans*, *Acetobacter_aceti*, *Arcobacter_butzleri*, *Azomonas_agilis*, *Lactobacillus_hordei* were extracted (Figure 3b).

### 3.4. To Elucidate Specific Bacteria That Reduce the Iron Correlation

Red blood cells can accumulate in OE and the pelvic cavity during retrograde menstruation. Hb, heme, and free iron are released as these cells break down. Based on the previous report, it was observed that the levels of total iron, heme, and free iron in EAOC were significantly lower in comparison to OE. Specifically, the concentrations were found to be 17 times lower for total iron, 11 times lower for heme, and 3 times lower for free iron [16]. In addition, the amount of iron in the liver can be accurately measured by analyzing the transverse magnetic relaxation rate R2 or R2* value through a complex, chemical shift-encoded MR examination. [13]. We reported that MR relaxometry in the cyst fluid could correlate with cyst fluid iron concentration [23,24]. We hypothesize that a particular type of bacteria may impact iron levels in cyst fluid, leading to a negative correlation between the proportion of bacterial copy number and the R2 value. A multiple regression analysis revealed the proportion of copy number of the genus *Bacillus* correlated with the R2 value in the combination of OE and EAOC samples (*p* = 0.007). Restricted in the OE samples, the proportion of the copy number of the genus *Bacillus* and *Propionibacterium* showed a significant correlation with the R2 value (*p* = 0.006 and *p* = 0.045) (Figure 4a). Moreover, the species *Lactobacillus_perolens* was extracted as an independent bacterium to correlate with the R2 value (*p* = 0.006). And restricted in the OE samples, the proportion of copy number of the *Acetobacter_aceti* was revealed to be an independent species to correlate with the R2 value (*p* = 0.030) (Figure 4b).

## 4. Discussion

To the best of our knowledge, this report represents the first attempt to identify factors that may promote malignant transformation by microbacteria in EAOC cyst fluids compared to normal OE cysts. It is well-known that iron-induced oxidative stress can cause DNA damage and regulate ROS metabolism through the NRF2 transcription factor, resulting in a higher risk of carcinogenic transformation due to mutation accumulation [25,26,27,28]. Several bacteria were shown to affect iron dynamics. For example, siderophores allow bacteria to solubilize iron from the environment and promote cell growth [29,30]. Uropathogenic Escherichia coli (UPEC) requires iron for growth. The ferritinophagy pathway contributes to its survival and may be involved in CDKN2A methylation [30,31]. Although a retrospective study, we observed that iron concentrations in EAOC are lower than in OE [16]. We also discovered a specific bacterial flora that is negatively correlated with iron concentrations. This flora may play a crucial role in reducing the formation of ROS species in iron-rich cyst fluid, which could promote malignant transformation.

Consistent with previous reports, bacteria belonging to the *Propionibacterium* genus were elucidated as the most outstanding macrobacterium between EAOC and OE cyst fluids, however, showing a relatively lower expression rate between these cyst fluids than the following factors [32]. *Bacillus* and *Propionibacterium* at the genus level and *Lactobacillus_perolens* and *Acetobacter_aceti* at the species level were elucidated in the correlation with iron component concentrations. The genus *Bacillus* comprises numerous species that can be found in water and soil worldwide. A significant number of these species are capable of thriving in extreme environments characterized by high pH, low temperatures, high salinity, and high pressure [33,34]. A recent study found that *Bacillus coagulans* can protect the gut from inflammation and oxidative damage. This is achieved by regulating TLR4/MyD88/NF-κB and Nrf2 signaling pathways in the intestinal tract of rats [35]. Although *Bacillus coagulans* was not specifically selected as a species for this study, the results suggest that it plays an important role in opposing ROS induced by iron. Studies show that HPV infection can lead to changes in the levels of certain bacteria in the body. These changes can include an increase in the amount of *Prevotella*, *Bacillus*, *Anaerococcus*, *Sneathia*, *Megasphaera*, and *Streptococcus*. It is believed that these changes could be linked to the severity of Cervical Intraepithelial Neoplasia (CIN), with an increase in *Bacillus* and *Anaerococcus* and a decrease in *Gardnerella vaginalis* potentially indicating a greater severity of the condition [36]. There are some studies suggesting that *Propionibacterium acnes* contributes to the development of several diseases in the prostate [37], gastric cancer [38], and ovarian cancer via regulating the hedgehog signaling pathway [32]. In this study, it was suggested that the genus level of *Propionibacterium* may be responsible for converting the cyst fluid environment to an environment with a lower concentration of iron. However, at the species level, *Propionibacterium acnes* did not show any difference in its dominance between EAOC and OE samples. Although the microbacterium *Lactobacillus_perolens* has not been studied in humans, the *Lactobacillus* genus is commonly found in a healthy human vagina. The disruption of the vaginal ecosystem can lead to harmful pathogens and infections, including bacterial vaginosis, sexually transmitted infections, and vulvovaginal candidiasis. The study revealed that *Lactobacillus* is significantly more dominant in EAOC compared to OE cyst fluids at the genus level, and among the identified *Lactobacillus* species, *perolens*, *vini*, and *hordei* were found to be the most prevalent. Interestingly, *Lactobacillus_perolens* was found to play a crucial role in reducing the iron content of cyst fluid, showing a significant impact.

It is important to note some limitations of this study due to its retrospective nature, which introduces some bias. Therefore, caution should be exercised when interpreting these findings. As this was a preliminary study, the next phase of our research involves the collection of bacterial strains from various sources, including the intestinal tract, vagina, intrauterine area, ovarian tumors, and abdominal cavity. Our aim is to investigate how these strains affect iron dynamics and to determine the route of infection, as well as the detailed mechanism of carcinogenesis. Additionally, our findings did not definitively identify whether certain bacteria contribute to the reduction of iron concentrations or if they are responsible for carcinogenic effects. The pathway of bacterial infection also remains unclear.

## 5. Conclusions

The EAOC cyst fluid contains numerous genera and species, but the genus *Bacillus* or *Propionibacterium* and the species *Lactobacillus_perolens* or *Acetobacter_aceti* show a potential connection to OE carcinogenesis.

## Figures and Tables

**Figure 1 microorganisms-12-00538-f001:**
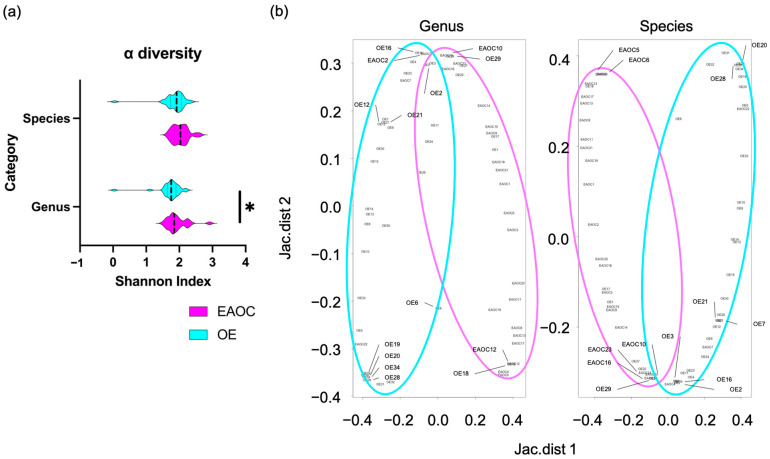
Violin plot of the Shannon index (**a**), and scatter plot of Jaccard index (**b**). Genus richness in EAOC is significantly higher than in OE. The asterisk * indicates significant differences (*p* < 0.05). The colors cyan and pink represent OE and EAOC, respectively. EAOC: endometriosis-associated ovarian cancer, OE: ovarian endometrioma.

**Figure 2 microorganisms-12-00538-f002:**
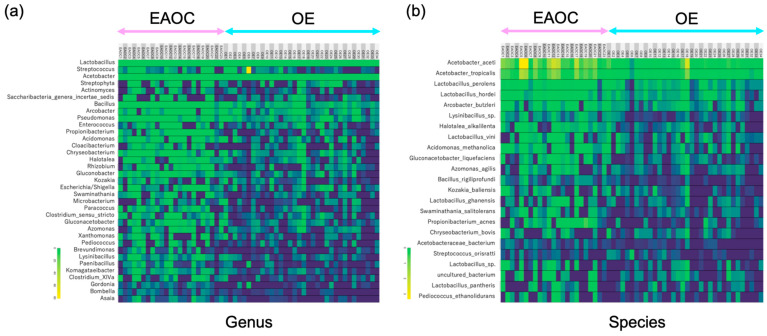
Heatmaps of genus (**a**) and species (**b**) among cyst fluid microbiota whose expression rate occupied over 50% of all detected bacteria. The colors cyan and pink represent OE and EAOC, respectively. EAOC: endometriosis-associated ovarian cancer, OE: ovarian endometrioma.

**Figure 3 microorganisms-12-00538-f003:**
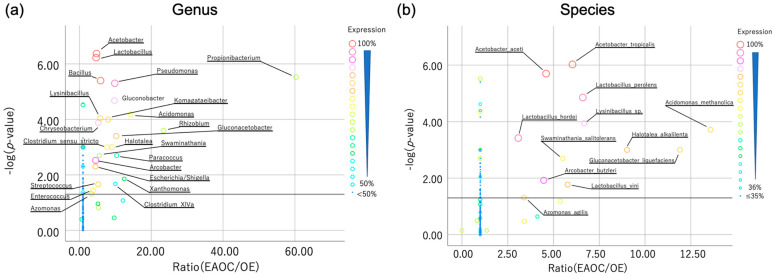
These are scatter plots that show the candidates related to malignant tumors of the genus (**a**) and species (**b**). The horizontal axis represents the expression rate of EAOC (endometriosis-associated ovarian cancer) in OE (ovarian endometrioma), while the vertical axis represents the logarithm of the *p*-value comparing the median expression of these groups. The circle on the plot represents the expression rate among all the samples. The horizontal line indicates a *p*-value of 0.05.

**Figure 4 microorganisms-12-00538-f004:**
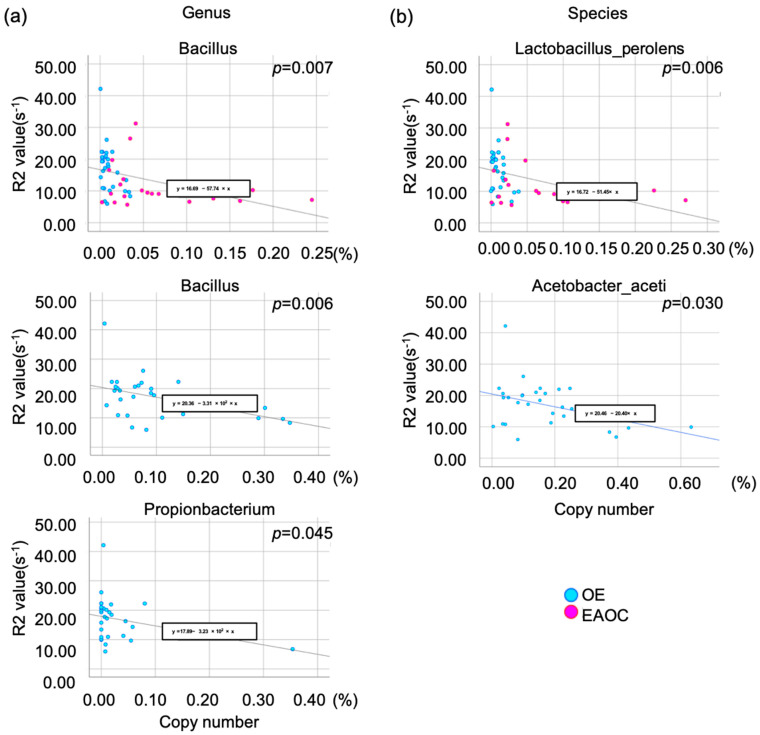
The correlation plots between micro-bacterial and R2 values suggest that the iron concentration of cyst fluids can be determined. In the genus analysis (**a**), the copy number proportion of *Bacillus* was found to be correlated with the R2 value (**upper**). When looking specifically at the OE samples, *Bacillus* (**middle**) and *Propionibacterium* (**lower**) showed a significant correlation with the R2 value. In the species analysis (**b**), *Lactobacillus_perolens* was identified as an independent bacterium that correlates with the R2 value in all samples (**upper**), while *Acetobacter_aceti* was extracted in the analysis of OE samples (**lower**).

**Table 1 microorganisms-12-00538-t001:** Demographic and clinical characteristics of the current cohort.

	0E	EAOC	*p*-Value
Number	*n* = 34	*n* = 23	
Age (years)			
Median (range)	39.00 (20–63)	52.00 (31–74)	
Mean ± SD	37.38 ± 9.50	50.57 ± 12.11	<0.001
BMI			
Median (range)	20.15 (16.02–26.72)	23.30 (17.00–28.90)	
Mean ± SD	20.59 ± 2.69	22.50 ± 3.79	0.104
Parity			
0	20	10	
≥1	14	13	0.193
Tumor volume (cm^3^)			
Median (range)	121.78 (5.71–637.94)	688.84 (44.71–2493.19)	
Mean ± SD	176.38 ± 154.51	824.67 ± 709.78	<0.001
R2 value (s^−1^)	^*1^	^*2^	
Median (range)	17.71(5.94–42.14)	9.12 (5.66–31.22)	
Mean ± SD	17.05 ± 7.27	11.58 ± 6.92	0.002

^*1^ five cases missing. ^*2^ three cases missing.

## Data Availability

Data are contained within the article.

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
