# Peer review of "Endogenous Microbacteria Can Contribute to Ovarian Carcinogenesis by Reducing Iron Concentration in Cysts: A Pilot Study"

_microorganisms, 2024, doi:10.3390/microorganisms12030538_

Round 1
Reviewer 1 Report
Comments and Suggestions for Authors
The present study elucidates the role of bacterial flora in endometriosis-associated ovarian cancer, specifically focusing on clear cell carcinoma. Employing next-generation sequencing, the investigation reveals the microbial composition within EAOC and its origin in ovarian endometriomas. A significantly higher Shannon index at the genus level is observed in EAOC compared to OE fluids, indicating a more diverse microbial community in cancerous environments. The findings highlight several bacterial species that are enriched in EAOC relative to benign chocolate cysts and demonstrate a strong correlation with decreased iron concentrations within the cysts. This observation suggests a potential role of these bacterial species in modulating iron levels and contributing to cancer development. Moreover, the identification of specific bacterial species associated with decreased iron concentrations in EAOC cysts sheds light on potential mechanisms of carcinogenesis in this context. The presence of diverse bacterial flora, particularly genera such as Bacillus and Propionibacterium, and species like Lactobacillus perolens or Acetobacter aceti, underscores the complex interplay between microbial communities and the ovarian microenvironment in promoting cancer progression. Understanding these interactions could pave the way for innovative therapeutic interventions targeting the microbiome to prevent or treat EAOC. The study provides novel insights into the role of bacterial flora in EAOC and warrants publication subject to minor grammatical corrections and inclusion of several references like Lancet, 2004, 364, 1789; Gynecol Oncol, 2006, 101, 331; Biol Chem. 2017, 398, 995; Mol Med 2021, 27, 33, etc.
Comments on the Quality of English LanguageMinor editorial corrections are required
Author Response
Thank you for your review and essential suggestions. We have incorporated some of the references in our study's discussion. This led to a suggestive discussion of the effects of bacteria on iron dynamics and reactive oxygen species metabolism. We also recognize that this is a pilot study and that further validation is essential.
Reviewer 2 Report
Comments and Suggestions for Authors
The study entitled 'Endogenous microbacteria can contribute to ovarian carcinogenesis by reducing iron concentration in cysts' investigates the potential role of bacterial flora in the development of endometriosis-associated ovarian cancer (EAOC), with a particular focus on changes in iron concentration within ovarian cysts. Although the study presents interesting findings, there are several critical points that require further scrutiny and discussion.
The study contributes to the growing body of literature exploring the microbiome's role in cancer development, specifically in ovarian carcinogenesis and iron metabolism.
However, the study design has limitations that need to be addressed. The research is retrospective, which introduces inherent biases that may impact the reliability of the findings. The study lacks experimental validation to establish causality between identified bacterial species and carcinogenic effects or iron concentration alterations. Future research should incorporate longitudinal or interventional studies to elucidate causal relationships.
Additionally, the study suggests a potential link between specific bacterial general species and ovarian carcinogenesis via iron concentration modulation. Bacillus, Propionibacterium, Lactobacillus_perolens, and Acetobacter_aceti are implicated in iron reduction within cyst fluid, potentially promoting malignant transformation. However, it is important to exercise caution when interpreting these findings. Although correlations between bacterial composition and iron concentration have been observed, the precise mechanisms by which these bacteria influence iron levels remain speculative.
Minor editing of English language required
Author Response
Thank you for taking the time to review our work and providing valuable suggestions.
We agree with your observation that our study is retrospective in nature; it is impossible to eliminate bias completely. Therefore, we have made changes to the title to highlight this aspect for the benefit of our readers. Additionally, we have mentioned our plans for further validation in the discussion section. Thank you again for your feedback and suggestions.
Round 2
Reviewer 2 Report
Comments and Suggestions for Authors
The current version of the manuscript is suitable for publication in 'Microorganisms'.